# L-Ascorbic Acid Restricts *Vibrio cholerae* Survival in Various Growth Conditions

**DOI:** 10.3390/microorganisms12030492

**Published:** 2024-02-29

**Authors:** Himanshu Sen, Manpreet Kaur, Saumya Ray Chaudhuri

**Affiliations:** 1CSIR-Institute of Microbial Technology, Sector 39A, Chandigarh 160036, India; chand0705@gmail.com (H.S.); manpreetkaur7009@gmail.com (M.K.); 2Academy of Scientific and Innovative Research (AcSIR), Ghaziabad 201002, India

**Keywords:** *Vibrio cholerae*, L-ascorbic acid, bile salts, biofilm, acetoin

## Abstract

Cholera, a deadly diarrheal disease, continues to ravage various parts of the world. It is caused by *Vibrio cholerae*, an important member of the gamma-proteobacteria. Based on certain genetic and phenotypic tests, the organism is classified into two major biotypes, namely classical and El Tor. The El Tor and its variants are majorly responsible for the ongoing seventh pandemic across the globe. Previously, we have shown that cross-feeding of glucose metabolic acidic by-products of gut commensals can severely affect the viability of the biotypes. In this work, we examined the effect of L-ascorbic acid on the survival of *Vibrio cholerae* strains belonging to both biotypes and different serotypes. We observed that L-ascorbic acid effectively restricts the growth of all strains under various conditions including strains adapted to acid stress. In addition, L-ascorbic acid is also effective in decreasing bile-induced biofilms of *Vibrio cholerae*.

## 1. Introduction

Cholera, a dreadful diarrheal disease of global importance, is caused by *Vibrio cholerae.* Since its discovery in the 19th century, the organism has been continuously studied worldwide. Over the years, extensive and intensive research has resulted in a wealth of information on the overall biology of the organism with a strong emphasis on its pathogenesis, mode of transmission, survival in hosts and diverse aquatic environments, and epidemiology [1]. Until now, 220 serogroups of *V. cholerae* strains have been reported in the literature, of which strains belonging to serogroups O1 and O139 are responsible for cholera epidemics [2,3]. Other than the O1 and O139 serogroups, the strains that fall into the remaining serogroups are clustered as non-O1 and non-O139, which are evidenced to cause local outbreaks [4,5]. It should be noted that the O1 serogroup of *V. cholerae* strains are further classified into two major biotypes, classical and El Tor. As per medical records, the world has experienced seven pandemics of cholera. The first six pandemics were caused by the classical biotype of O1 serogroup strains, whereas the El Tor biotype of O1 serogroup strains is responsible for the ongoing seventh pandemic around the world [6]. Although both the classical and El Tor biotypes are closely related, some genetic and biochemical differences are still present between the two biotypes including a unique difference in carbohydrate metabolism. It is documented that the viability of classical biotype strains (e.g., O395) is drastically affected in the presence of glucose due to the production of organic acids, whereas the El Tor strains (e.g., N16961) evolved with the machinery to convert glucose into acetoin, a neutral fermentation end product that promotes the growth of El Tor strains [7]. This evolutionary fitness is believed to be one of the contributing factors to the supremacy of El Tor strains over classical strains in the ongoing seventh pandemic, which started way back in 1961 [8].

Among multiple lines of treatment, oral or intravenous rehydration remains the mainstay in cholera. Antibiotics are also given as an adjunct therapy to control and decrease the duration of the disease. However, indiscriminate use of antibiotics increases the emergence of multidrug-resistant strains of *V. cholerae* [5,9], thereby contributing to the global burden of AMR. Vaccines with limited efficacy providing short-term protection are also available and can be useful to restrict the spread of cholera in outbreak-prone areas. Presently, there are three WHO-approved oral cholera vaccines (OCVs), namely Dukoral, Shankol, and Evichol. These vaccines require two doses for full protection. Studies on mass vaccination with Dukoral and Shankol have clearly demonstrated the safety of these vaccines for pregnant women. However, these OCVs provide limited protection among children under 5 years and none of these OCVs is recommended to infants. The current strategies are oriented to develop single-dose immunization over two-dose regimens; improved formulation to provide protection among young children; and increased production levels to supply OCVs in outbreak regions. Furthermore, current vaccines can give protection against cholera for 6 months up to 3 years [10,11,12,13]. Therefore, there is a desperate need to come up with newer strategies to combat cholera. Interestingly, recent studies have clearly demonstrated that feeding acidic metabolites resulting from the sugar fermentation of gut commensal and probiotics strains is an effective way to restrict the growth and pathogenesis of both classical and El Tor strains in in vitro and animal models [14,15,16].

Since its discovery in 1920, vitamin C has been shown to be an antibacterial agent against a number of pathogenic bacteria [17]. The molecule also exerts immunomodulatory activity at high concentrations [18]. Recently, sodium salt of vitamin C (sodium ascorbate) has been evidenced to act as a quorum-sensing inhibitor in *V. campbellii* [19].

To cause cholera, *V. cholerae* must effectively colonize the small intestine. Before reaching the small intestine, the pathogen must travel through the acidic barrier in the stomach. To circumvent and survive such a hostile environment, *V. cholerae* mounts a powerful acid tolerance response (ATR). The contribution of ATR to the infectious life cycle of *V. cholerae* is well recognized [20,21]. Once it passes through the stomach acidic barrier, *V. cholerae* encounters bile in the proximal duodenal lumen. Due to its detergent-like properties, bile is important for solubilizing lipids during digestion, and it is also known to possess bactericidal activity [22,23]. Even though bile is highly effective in reducing the *V. cholerae* population in the gut, it also promotes robust biofilm formation by the pathogen [22]. Such exploitation of bile, an important physiologic component of the intestine, also contributes to cholera pathogenesis.

In this work, we examined the effect of L-ascorbic acid (L-AA) on the growth of various strains of *V. cholerae* under various growth conditions. Our data clearly indicated that L-AA effectively restricts the growth of *V. cholerae* strains irrespective of the biotype and serotype examined in the present study. We also demonstrated growth mitigation in acid-adapted *V. cholerae* strains in the presence of L-AA. Furthermore, L-AA effectively reduces the cell viability of the bile-induced biofilm of the pathogen.

## 2. Materials and Methods

### 2.1. Bacterial Strains and Media

All bacterial strains used for this study are listed in Table 1. All *V. cholerae* strains were grown in Lysogeny broth (LB) at 37 °C or on solid agar. Streptomycin was used at a concentration of 0.1 mg/mL wherever needed. All medium ingredients were purchased from BD Difco (Franklin Lakes, NJ, USA) and salts were procured from Sigma-Aldrich (St. Louis, MO, USA). Bile salts and Barritt reagents A and B were purchased from Himedia (Mumbai, India).

### 2.2. Growth Assays

A liquid growth assay was set up by diluting the early log phase cultures of *V. cholerae* strains to an optical density (OD) 600_nm_ of 0.01 in LB supplemented with salts at 37 °C. OD 600_nm_ was measured at regular intervals for 10 h. The pH of the cultures was measured.

For viability spotting assays, the liquid cultures were set up as mentioned above. After the stipulated period, the cultures were serially diluted in phosphate-buffered saline (PBS) and spotted on LB agar. Plates were photographed after 16 h of growth at 37 °C.

### 2.3. Acid Tolerance Response (ATR) and Viability Assays in the Presence of L-Ascorbic Acid

ATR and viability of *V. cholerae* assays were performed as previously described with slight modifications [20,26]. The overnight-cultured *V. cholerae* strains were diluted into LB medium and incubated at 37 °C until they reached an OD 600_nm_ of 0.4. The strains were divided into two vials, one with 10% and the other with 90% of the culture. Cells were then washed with LB and then the 10% cells were resuspended in LB pH 7.0 and 90% cells in LB pH 5.7 (pH was adjusted using 1N HCl or L-AA) and incubated at 37 °C for 2 h. Subsequently, the strains cultured in pH 7.0 or pH 5.7 LB medium were transferred into LB medium at pH 4.5 (adjusted using either HCl or L-AA). An aliquot was taken from each culture at the indicated time points, diluted appropriately, and spotted on LB agar plates.

### 2.4. Bile-Induced Biofilm Assay and Viability in the Presence of L-Ascorbic Acid

The bile-induced biofilm was set up as previously described with minor alterations [22]. *V. cholerae* strains were grown overnight in LB. Overnight cultures were resuspended in LB and grown till they reached an OD 600_nm_ of 0.4. These cultures were then used in a 1:100 dilution to inoculate LB broth or LB + 0.4% bile salts in borosilicate glass tubes and incubated at 22 °C for 22 h. The next day, the liquid cultures carrying planktonic cells were removed, and OD 600_nm_ was measured. Meanwhile, the tubes were washed with PBS twice and stained with 0.1% crystal violet for 15 min. Stained biofilms were again washed twice with PBS. The tubes were then photographed. Crystal violet stain was dissolved in dimethyl sulphoxide (DMSO), and the OD 570_nm_ was measured. Student’s *t*-test was performed to measure the *p*-value and statistical significance.

To check the biofilm-incorporated cell viability, biofilms with bile salts were set up as described previously. After the stipulated incubation time, biofilms were washed with PBS and treated with LB or LB + 0.4% bile salts. Biofilms were then treated with L-AA (5 mg/mL) for 3 h and cell viability was checked by resuspending the adhered biofilm in PBS using sterile cotton swabs; serial dilutions were then prepared and spotted on LB agar. Plates were photographed after 16 h of growth at 37 °C.

### 2.5. Voges–Proskauer Test

The Voges–Proskauer (VP) test was performed as described earlier with some modifications [27]. Briefly, the *V. cholerae* strains N16961 and C6706 were grown in LB or LB supplemented with 1% glucose (LBG). L-AA was added to the LBG media flasks at either 0 h or 10 h of growth. 0.1 mL volume of each culture was mixed with 0.02 mL of Barritt Reagent A and 0.01 mL of Barritt Reagent B.

## 3. Results

### 3.1. Viability of V. cholerae Strains Is Severely Compromised in the Presence of L-Ascorbic Acid

To ascertain the survival of *V. cholerae* strains, *V. cholerae* classical (e.g., O395), El Tor (e.g., C6706), and non-O1/non-O139 strains (e.g., PL91) were subjected to growth assays in LB containing different concentrations of L-AA. All strains exhibited severe growth reduction in the liquid growth assay in the presence of the molecule at concentrations of 2.5 mg/mL and 5 mg/mL (Figure 1A). The viability spotting also corroborated the same (Figure 1B). In addition to the *V. cholerae* strains O395, C6706, and PL91, the effect of L-AA was examined on more *V. cholerae* strains (e.g., El Tor atypical, and non-O1/non-O139, Table 1) and observed similar growth inhibition by the molecule (Appendix A). It should be noted that the addition of L-AA caused extreme acidification of the growth medium, which seemed responsible for the decimation of *V. cholerae* strains.

### 3.2. L-Ascorbic Acid Mitigates the Growth of Acid-Adapted V. cholerae Strains

*V. cholerae* is sensitive to acidic pH and most is killed by stomach acid [28]. Previous studies show that *V. cholerae* strains exposed to acid stress can survive at low pH [20]. Acid adaptation promotes intestinal colonization of the bacterium [29]. We asked whether L-AA can affect the growth of acid stress-adapted *V. cholerae*. To investigate, *V. cholerae* strains C6706 and N16961 were subjected to an acid adaptation using HCl as per the published protocol [20]. Both unadapted and adapted strains were subsequently exposed to low-pH LB media (henceforth LLB pH-4.5) where pH was adjusted either using 1 N HCl (LLB-HCl) or with L-ascorbic acid (LLB-LA). As expected, unadapted strains did not survive in both types of LLB media. Interestingly, the adapted strains exhibited much better killing in LLB-LA medium than in the LLB-HCl medium (Figure 2A). Likewise, we adapted both strains with L-AA and exposed them to LLB-HCl and LLB-LA. In this case, both strains failed to survive in LLB-LA and LLB-HCl and showed no acid adaptation in the presence of L-ascorbic acid (Figure 2B).

### 3.3. L-Ascorbic Acid Affects Bile-Induced Biofilm Growth of Vibrio cholerae Strains

After the stomach acid, *V. cholerae* is exposed to bile in the proximal duodenal lumen. Bile exerts antimicrobial activity and is an important component of the intestinal defense system [23]. Surprisingly, bile acids promote biofilm formation in *V. cholerae*. Bile acid-mediated induction of biofilm development involves the transcriptional regulator VpsR. It has been demonstrated that bile acid-treated cultures contain 2.5-fold more cells in the biofilm than LB control, and such bile acid-induced biofilm formation protects *V. cholerae* from the bactericidal effect of bile [22]. We wanted to investigate whether L-AA could affect the viability of bile-induced biofilms in *V. cholerae*. To this end, a biofilm was developed for 22 h with 0.4% bile acids as per the published protocol, and both the *V. cholerae* strains N16961 and C6706 showed enhanced biofilm formation (Figure 3A) [22]. L-AA was administered exogenously to the mature biofilm, and cell viability was measured. We observed a drop in pH followed by significant growth reduction within the biofilm in the presence of L-AA (Figure 3B).

### 3.4. L-Ascorbic Acid-Mediated Acidification in Growth Medium Containing ORS Salt

As described in the preceding sections, L-AA is quite effective in controlling the growth of various *V. cholerae* strains under various physiological conditions. Therefore, we wanted to examine the efficacy of the L-AA in the presence of glucose and ORS salt, the major constituents of oral rehydration solution. It is noteworthy to mention that some *V. cholerae* strains (e.g., N16961) convert glucose into acetoin (a neutral compound that also serves as an energy source) and gain an advantage on growth in culture media containing glucose, while other *V. cholerae* strains (e.g., O395) produce acidic metabolites as a result of glucose fermentation and encounter a severe challenge in surviving under such conditions [7,14,15]. The conversion of glucose to acetoin might account for the evolutionary fitness of *V. cholerae* El Tor biotypes (e.g., N16961) over *V. cholerae* classical biotypes (e.g., O395) [7]. Keeping this view in mind, we wanted to check whether L-AA can tackle acetoin-producing *V. cholerae* strains in a glucose-containing growth medium. To pursue our interest, acetoin-positive *V. cholerae* El Tor strains N16961 and C6706 were subjected to growth in LB containing 1% glucose and L-AA (5 mg/mL) as this concentration was more effective in restricting the growth (Figure 1B) in two ways. In one case, strains were inoculated in LB containing both glucose and L-AA, and in another case, strains were grown for 10 h in LB containing glucose to produce acetoin (Appendix A), followed by challenge with L-AA (5, 10 and 20 mg/mL) and growing for an additional 6 h. After 16 h of incubation, the viability was checked by spotting on LB agar. We observed the killing of both strains under these conditions, and pH was found to be acidic at the end of the incubation (Figure 4A). This further indicates that L-AA is able to overcome acetoin-mediated growth advantage and can lower the pH, thus affecting the survival of these strains.

To underscore the effect of ORS salts on L-AA-mediated growth retardation, *V. cholerae* strains N16961 and C6706 were grown in LB containing a mixture of salts present in ORS, and L-AA was added in varying concentrations at different phases of growth. The viability spotting data clearly demonstrated that a higher concentration of L-AA (20 mg/mL) is effective in causing complete growth cessation of both strains at the stationary phase (Figure 4B).

Recently, *V. cholerae* has been shown to utilize L-ascorbate as an energy source. Interestingly, Boyd and colleagues grew *V. cholerae* in M9CAS medium containing glucose or L-AA [30]. On the other hand, we carried out our growth experiments in LB medium. We postulate that M9CAS salt medium neutralizes the acidic conditions following the addition of L-ascorbate, thus promoting the growth of the bacteria, and the situation further enables *V. cholerae* to utilize L-ascorbate. In order to examine this, we grew *V. cholerae* strains in M9CAS medium containing either glucose or L-ascorbic acid at concentrations of 5 mg/mL and 10 mg/mL. No killing was observed at this concentration of L-AA and the pH was not acidic either (Figure 4C). Therefore, we increased the concentration of L-AA to 20 mg/mL and 50 mg/mL. Both these concentrations of L-AA severely mitigated the growth of *V. cholerae* strains (Figure 4C).

Collectively, we have demonstrated the efficacy of L-AA in restricting the growth of different *V. cholerae* strains under various growth conditions.

## 4. Discussion

The emergence of drug-resistant *Vibrio cholerae* El Tor strains remains the biggest factor in the seventh pandemic of cholera that has been ongoing for over six decades now [8]. To tackle this serious threat, alternative strategies are being continuously discovered in parallel to the existing line of treatments. For example, small molecules like virstatin have been shown to reduce the virulence of *V. cholerae* by directly inhibiting ToxT, a master regulatory protein that modulates the production of primary virulence factors (e.g., cholera toxin) [31,32]. However, natural variants of ToxT have also been identified and found resistant to virstatin-mediated functional modulation [33]. Quorum-sensing inhibitors are also considered a potential substitute for antibiotics. Small molecules targeting master regulatory proteins such as LuxO and LuxR/HapR have been shown to be effective in reducing *Vibrio* pathogenesis [34,35,36,37]. Other than quorum-sensing master regulators, proteins contributing to the biofilm formation of *Vibrio cholerae* have also been targeted for small molecule-mediated inhibition [38,39,40,41]. The sensitivity of *V. cholerae* to acidic environments can also be an attractive target and should be exploited further to control cholera pathogenesis. In fact, the introduction of glucose-containing ORS is now believed to eliminate the glucose-sensitive *Vibrio cholerae* classical biotype [7]. Even to tackle the threat posed by the glucose-resistant *V. cholerae* El Tor strains, cross-feeding of acidic metabolites produced from the glucose metabolism of commensal probiotic strains has proven effective [14,15,16]. Moreover, small molecule-mediated functional perturbation of the metabolic pathway that converts glucose to neutral molecules also makes glucose-resistant El Tor biotypes susceptible to acid stress [42]. In short, creating an acidic environment might control the growth of the organism.

In the current work, we wanted to examine the effect of L-ascorbic acid, a vitamin, a natural antibiotic, and an acidifier that shows antimicrobial activity on the growth of *V. cholerae* strains under various growth conditions. Our results clearly indicated that L-ascorbic acid at concentrations as low as 2.5 mg/mL is highly toxic to the pathogen (Figure 1, Figure 2 and Figure 3). This result is in accordance with the effects of L-AA shown previously on *Pseudomonas aeruginosa* and uropathogenic *E. coli* [43,44]. It is noteworthy to mention that alkaline pH not only favors the growth of *Vibrio cholerae* but also increases its mucus penetrability [45]. Our data documented the efficacy of L-AA in bringing down the alkaline pH to the inhabitable acidic range and affecting *V. cholerae* growth (Figure 4A,B). Moreover, both hydrochloric acid and L-AA-adapted *V. cholerae* strains failed to grow in the presence of L-AA, indicating the role played by the antibacterial activity of L-ascorbic acid besides its acidic pH (Figure 3A,B).

The benefits and safety of ascorbic acid are well recognized and established (https://www.chemicalsafetyfacts.org/chemicals/ascorbic-acid/, accessed on 23 August 2023). A recent study also suggested the beneficial effect of vitamin C on gut health [46]. It should be noted that *V. cholerae* invasion causes a tremendous impact on the gut community and some members of the gut microbiota play a key role in the cholera recovery phase to restore community structure and function [47,48]. It would be interesting to examine the effect of the combination of ORS and L-AA in the in vivo growth mitigation of *V. cholerae* and restoration of the gut microbiota after diarrhea. Additional studies are necessary to address these issues.

## Figures and Tables

**Figure 1 microorganisms-12-00492-f001:**
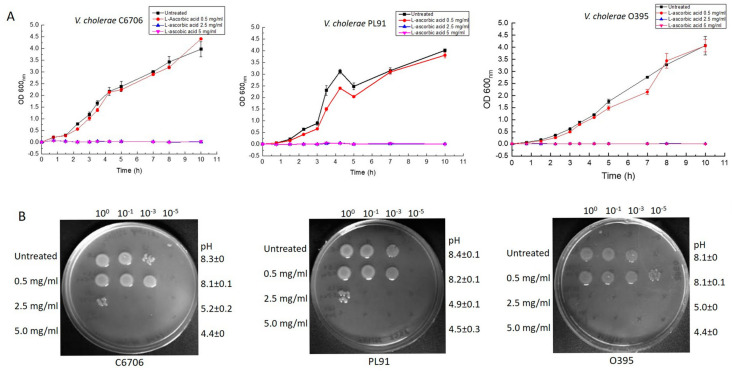
Effect of L-ascorbic acid on *Vibrio cholerae* growth. *Vibrio cholerae* strains C6706, PL91, and O395 were either grown in LB or LB supplemented with increasing concentrations of L-ascorbic acid (0.5 mg/mL, 2.5 mg/mL, and 5 mg/mL). (**A**) Log-phase growth cultures were diluted to a starting OD 600_nm_ of 0.01 and OD 600_nm_ was measured at regular intervals. Error bars indicate standard deviation from the mean calculated using the values obtained from biological and technical duplicates of the experiment. (**B**) Cultures grown as in (**A**) were serially diluted and spotted on solid agar plates. Plates were photographed after 16 h of incubation at 37 °C. The plate photographs are representative of the experiment performed in biological and technical duplicates. pH values were measured at the end of the growth assay and standard deviation from the mean was calculated. pH values and their standard deviations were rounded off and reported.

**Figure 2 microorganisms-12-00492-f002:**
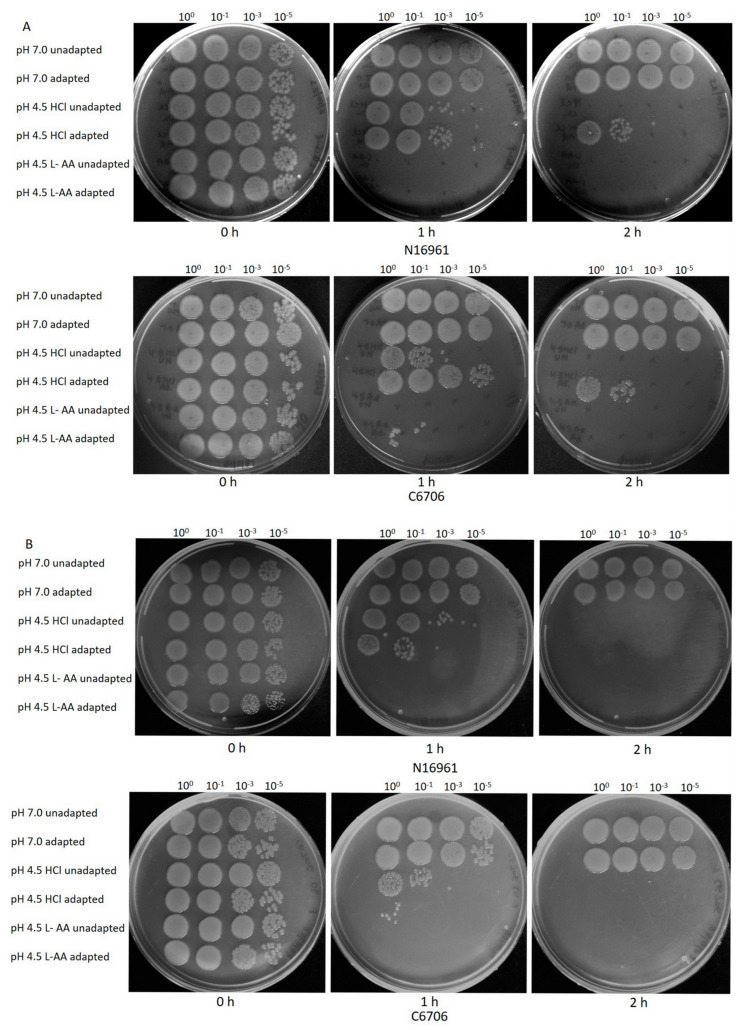
L-AA accelerates killing of acid-adapted *V. cholerae* El Tor strains. Overnight cultures of *V. cholerae* strains N16961 and C6706 were diluted 1:100 in LB and grown till early log phase. Cultures were then divided in the ratio of 1:9 with one part grown in LB pH 7.0 (unadapted) and nine parts being used for adaptation in LB pH 5.7 (adjusted using either (**A**) HCl or (**B**) L-ascorbic acid) for 2 h. Cultures were then subjected to acid shock in LB at pH 4.5 where pH was adjusted either with HCl or L-ascorbic acid. Serial dilutions were spotted on LB agar plates at time intervals of 0 h, 1 h, and 2 h. Plates were photographed after 16 h of incubation at 37 °C.

**Figure 3 microorganisms-12-00492-f003:**
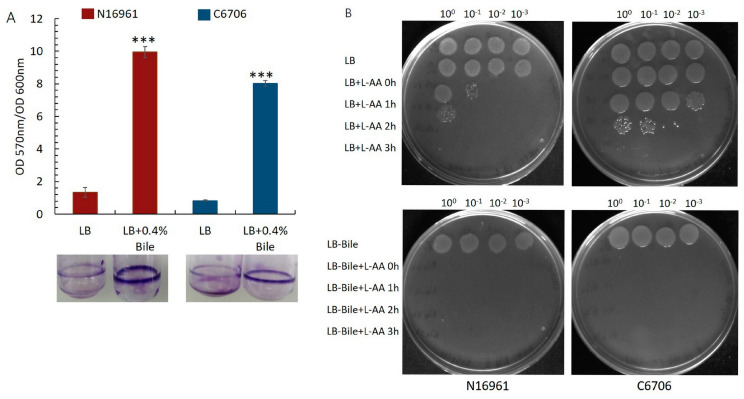
Effect of L-AA on the bile salt-induced biofilm formation in *V. cholerae.* Biofilms of strains N16961 and C6706 were grown in control LB media and LB + 0.4% bile salts without shaking at 22 °C for 22 h. After the incubation time, the planktonic cells were removed and the biofilm ring was washed with 1X PBS. (**A**) Biofilm was stained with 0.1% crystal violet for 15 min and washed twice with 1X PBS. Pictures of the stained tubes were taken. Crystal violet was dissolved in DMSO and absorbance at 570_nm_ was measured. The ratio of OD570_nm_/OD600_nm_ was then plotted as a bar graph. Standard deviations were obtained from biological replicates. Student’s *t*-test was performed and *** indicates *p* value <  0.001. (**B**) Pre-formed biofilms were either treated with LB + L-AA 5 mg/mL or LB + 0.4% bile + L-AA 5 mg/mL for 3 h. Culture viability was checked at regular time intervals by dissolving the biofilms using a sterile cotton swab and spotting the serial dilutions of the biofilm cultures on LB agar plates. Plates were incubated at 37 °C for 16 h. Plates shown are the best representatives of the experiment performed in replicates.

**Figure 4 microorganisms-12-00492-f004:**
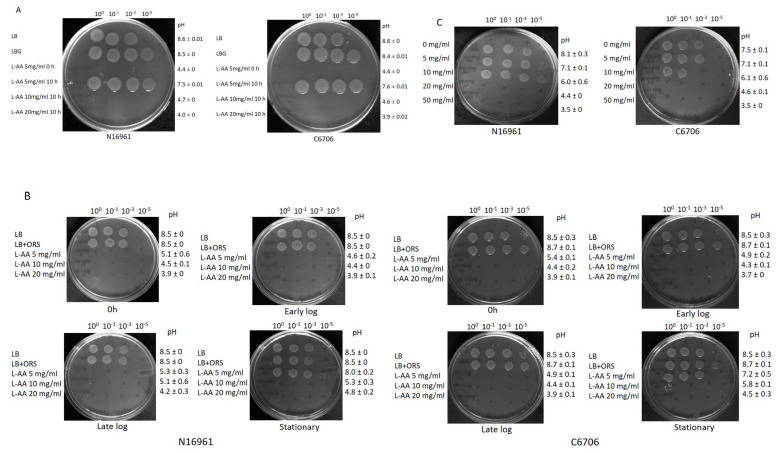
Effect of L-ascorbic acid on *V. cholerae* El Tor strains grown in the presence of glucose and various salts. Log-phase growth cultures of *V. cholerae* strains N16961 and C6706 were diluted to a starting OD 600_nm_ of 0.01 and (**A**) were grown in either LB or LB + 1% glucose (LBG). L-AA (5 mg/mL) was added to the LBG flask at the start of the experiment and after 10 h of growth at a concentration of 5, 10, or 20 mg/mL. (**B**) *V. cholerae* cultures were grown in LB supplemented with the oral rehydration salt mix (ORS). Cultures were challenged with varying concentrations of L-AA (5, 10, and 20 mg/mL) at different growth phases [i.e., at the beginning of the assay (0 h) and the early log phase, late log phase, and stationary phase of the cultures]. (**C**) Both El Tor strains of *V. cholerae* were grown in M9 minimal media supplemented with 20 mM casamino acids and increasing concentrations of L-AA (0, 5, 10, 20, and 50 mg/mL). For (**A**–**C**), cultures were serially diluted and spotted on solid agar. Plates were photographed after 16 h of incubation at 37 °C. The plates shown are representative of the experiment performed in replicates. The pH values were measured at the end of each assay and the standard deviation from the mean was calculated and rounded off.

**Table 1 microorganisms-12-00492-t001:** *Vibrio cholerae* strains used in this study.

Strain	Genotype	Source
C6706	O1, El Tor, variant, streptomycin-resistant (Sm^R^)	Ron Taylor, Dartmouth Geisel School of Medicine USA
PL91	Non-O1, non-O139, Serogroup O110,	[24]
O395	O1, classical	Andrew Camilli, Tuft University
N16961	O1, El Tor, Ogawa, Sm^R^	Andrew Camilli, Tuft University
SC110	Non-O1, non-O139 serogroup O34	[25]
014-99	El Tor variant	Richard Y.C. Kong, City University of Hong Kong

## Data Availability

Data are contained within the article.

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
