# Peer review of "L-Ascorbic Acid Restricts Vibrio cholerae Survival in Various Growth Conditions"

_microorganisms, 2024, doi:10.3390/microorganisms12030492_

Round 1
Reviewer 1 Report
Comments and Suggestions for Authors
This paper aims to study the effect of L-ascorbic acid on the survival of Vibrio cholerae strains belonging to both biotypes and different serotypes.
The manuscript is written comprehensively enough to be understandable; the authors addressed this aim by demonstrating the mode of action of V. cholerae and showcasing the role of L-ascorbic acid (L-AA) on the growth of diverse strains of V. cholerae under various growth conditions and how it effectively restricts the growth of its strains irrespective of the biotype and serotype examined in this study. They also elucidated that Vit C effectively reduces cell viability of the bile-induced biofilm of the pathogen.
The paper stated the purpose, discussion and global implication are clearly stated and consistent with the rest of the manuscript; they addressed their hypothesis and opinion in a reproducible way and proved their results through all the required experiments and analysis and they used enough number of analyses to prove their results. The results were presented in a clear way which should facilitate in reaching a clear conclusion, however the authors did not succeed to make a conclusion for their study so I would suggest to make it clear and understandable.
To improve the introduction, I suggest that authors should talk more about Cholera’s vaccine, treatment, molecules in clinical trials, etc.
The abbreviations were explained at the first place they are mentioned.
In vitro, in vivo, et al.: should be written in italic.
No plagiarism has been detected.
References: The authors followed the journal guidelines for some references.
Comments on the Quality of English Language
No comments
Author Response
"Please see the attachment.

Reviewer 2 Report
Comments and Suggestions for Authors
The article entitled « L-Ascorbic acid restricts Vibrio chloerae survival in various growth conditions » focuses on the effect of L-Ascorbic Acid (L-AA)on different Vibrio chloerae, belonging to different Biotypes and serotypes, among which known some strains involved in serious diarrheal diseases. The authors demonstrated the inhibitory effect on the Vibrio strains growth of the L-AA used at 2.5 and 5 mg/ml, assigning the cause (already described) to a pH effect (acid) induced by the L-AA. The antimicrobial effect of L-AA was also highlighted on acid-adapted strains, corresponding to the adaptation of some acid-tolerant strains for intestinal colonizing The authors have studied the impact of L-AA on the bile-induced biofilm and showed a significant reduction of growth when the biofilm was treated with L-AA.
At least, L-AA was shown active in Glucose/ORS condition suggesting the capacity of L-AA to overcome the acetoin-mediated growth advantage for the Vibrio strains. The real role of L-AA was also confirmed by neutralizing L-AA (using M9CAS salt) and the observation of the Vibrio growth.
Comments :
The article is well- structured, with the objectives correctly exposed. However, some informations are lacking and lead the reader to misunderstanding. The interest and relevance of the context of this study have to be more clear (see comment). The discussion takes up the main results without a real discussion. This article can be accepted with major revisons (see comments below)
Comments :
1-The interest of the sutdy have to be clearly mentioned. Some information related to the potential of the use of L-AA in therapeutic treatment or other, is not indicated. Are there previous studies /articles mentioning the use and the mechanism of LAA ?
2-Materials and Methods must be completed :
-Lane 75-76 : « Sigma-Aldrich » and « Difco » : Indicate the country of the supplier.
-Table 1 : The format of the Table is not conventional (Avoid black or grey backgroud, no columns at the borders of the table…) ; Must be corrected.
-Table 1 : The abbreviation of « SmR » has to be identified in the caption. Which resistance ?
-Lane 108 : in biofilm section, the concentration of the L-AA used has to be indicated.
-Figure 3 : Some results are presented as « significant ». But no statistical tests are mentioned in the mat et meth. Indicate the test used.
3-Figue 2 : The indication of (A) and (B) in the caption (A and B are assoiciated to the strains) is not in line with the figure ( A and B seem to be associate with the result of duplicat).If it correspond to duplicate, it is not relevant to show the same thing. Do the correction/figure.
4-Lanes 119-140 :
-The choice of the 3 concentrations (O.5, 2.5 and 5 mg :ml) of L-AA is not argued. Why these concentrations ?
-Why no intermediate concentrations between 0.5 and 2.5 have been tested ?
-Is there a dose effect of L-AA on the growth ?
-L-AA is a bacteriocide or bacteristatic ?
5-Lane 128-129 : What is the behaviour of the pH during the growth (figure 1A) ? is it stable and acid ?
6-Lane 156-153 : It indicates that when acid pre-treatment is performed with L-AA, no acid adaptation has been observed (both for HCl and L-AA). It would mean that L-AA killed the microorganisms (because L-AA is bacteriscide ?), and there is an action of L-AA not involving acid parameter (because pre-treatment with HC lis not letal). Is there data about this action ? This point is not discussed in the discussion section. It should be discussed.
7-Discussion is poor. Must be improved in term of discussion of results
Author Response
"Please see the attachment.

Round 2
Reviewer 2 Report
Comments and Suggestions for Authors
The authors have improved the manuscript and answerd to the questions.
The new version of the manuscript can be published.